# Beyond Shared Hierarchies: Deep Multitask Learning through Soft Layer Ordering

**Elliot Meyerson & Risto Miikkulainen**
The University of Texas at Austin and Sentient Technologies, Inc.
{ekm, risto}@cs.utexas.edu

## Abstract

Existing deep multitask learning (MTL) approaches align layers shared between tasks in a *parallel ordering*. Such an organization significantly constricts the types of shared structure that can be learned. The necessity of parallel ordering for deep MTL is first tested by comparing it with *permuted ordering* of shared layers. The results indicate that a flexible ordering can enable more effective sharing, thus motivating the development of a *soft ordering* approach, which learns *how* shared layers are applied in different ways for different tasks. Deep MTL with soft ordering outperforms parallel ordering methods across a series of domains. These results suggest that the power of deep MTL comes from learning highly general building blocks that can be assembled to meet the demands of each task.

## 1 Introduction

In multitask learning (MTL) (Caruana, 1998), auxiliary data sets are harnessed to improve overall performance by exploiting regularities present across tasks. As deep learning has yielded state-of-the-art systems across a range of domains, there has been increased focus on developing deep MTL techniques. Such techniques have been applied across settings such as vision (Bilen and Vedaldi, 2016; 2017; Jou and Chang, 2016; Lu et al., 2017; Misra et al., 2016; Ranjan et al., 2016; Yang and Hospedales, 2017; Zhang et al., 2014), natural language (Collobert and Weston, 2008; Dong et al., 2015; Hashimoto et al., 2016; Liu et al., 2015a; Luong et al., 2016), speech (Huang et al., 2013; 2015; Seltzer and Droppo, 2013; Wu et al., 2015), and reinforcement learning (Devin et al., 2016; Fernando et al., 2017; Jaderberg et al., 2017; Rusu et al., 2016). Although they improve performance over single-task learning in these settings, these approaches have generally been constrained to joint training of relatively few and/or closely-related tasks.

On the other hand, from a perspective of Kolmogorov complexity, "transfer should always be useful"; any pair of distributions underlying a pair of tasks must have *something* in common (Mahmud, 2009; Mahmud and Ray, 2008). In principle, even tasks that are "superficially unrelated" such as those in vision and NLP can benefit from sharing (even without an *adaptor* task, such as image captioning). In other words, for a sufficiently expressive class of models, the inductive bias of requiring a model to fit multiple tasks simultaneously should encourage learning to converge to more *realistic* representations. The expressivity and success of deep models suggest they are ideal candidates for improvement via MTL. So, why have existing approaches to deep MTL been so restricted in scope?

MTL is based on the assumption that learned transformations can be shared across tasks. This paper identifies an additional implicit assumption underlying existing approaches to deep MTL: this sharing takes place through *parallel ordering* of layers. That is, sharing between tasks occurs only at aligned levels (layers) in the feature hierarchy implied by the model architecture. This constraint limits the kind of sharing that can occur between tasks. It requires subsequences of task feature hierarchies to match, which may be difficult to establish as tasks become plentiful and diverse.

This paper investigates whether parallel ordering of layers is necessary for deep MTL. As an alternative, it introduces methods that make deep MTL more flexible. First, existing approaches are reviewed in the context of their reliance on parallel ordering. Then, as a foil to parallel ordering, *permuted ordering* is introduced, in which shared layers are applied in different orders for different tasks. The increased ability of permuted ordering to support integration of information across tasks is analyzed, and the results are used to develop a *soft ordering* approach to deep MTL. In this

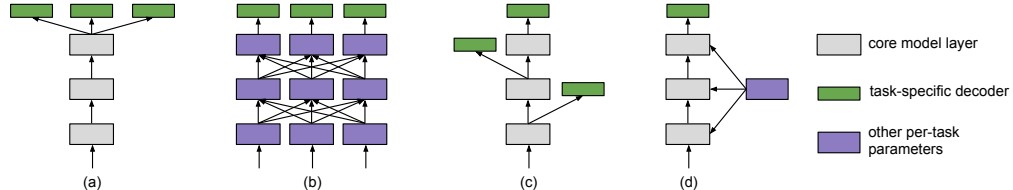

Figure 1: **Classes of existing deep multitask learning architectures.** (a) *Classical approaches* add a task-specific decoder to the output of the core single-task model for each task; (b) *Column-based approaches* include a network column for each task, and define a mechanism for sharing between columns; (c) *Supervision at custom depths* adds output decoders at depths based on a task hierarchy; (d) *Universal representations* adapts each layer with a small number of task-specific scaling parameters. Underlying each of these approaches is the assumption of parallel ordering of shared layers (Section 2.2): each one requires aligned sequences of feature extractors across tasks.

approach, a joint model learns *how* to apply shared layers in different ways at different depths for different tasks as it simultaneously learns the parameters of the layers themselves. In a suite of experiments, soft ordering is shown to improve performance over single-task learning as well as over fixed order deep MTL methods.

Importantly, soft ordering is not simply a technical improvement, but a new way of thinking about deep MTL. Learning a different soft ordering of layers for each task amounts to discovering a set of *generalizable modules that are assembled in different ways for different tasks*. This perspective points to future approaches that train a collection of layers on a set of training tasks, which can then be assembled in novel ways for future unseen tasks. Some of the most striking structural regularities observed in the natural, technological and sociological worlds are those that are repeatedly observed across settings and scales; they are ubiquitous and universal. By forcing shared transformations to occur at matching depths in hierarchical feature extraction, deep MTL falls short of capturing this sort of functional regularity. Soft ordering is thus a step towards enabling deep MTL to realize the diverse array of structural regularities found across complex tasks drawn from the real world.

## 2 PARALLEL ORDERING OF LAYERS IN DEEP MTL

This section presents a high-level classification of existing deep MTL approaches (Sec. 2.1) that is sufficient to expose the reliance of these approaches on the *parallel ordering assumption* (Sec. 2.2).

### 2.1 A CLASSIFICATION OF EXISTING APPROACHES TO DEEP MULTITASK LEARNING

Designing a deep MTL system requires answering the key question: *How should learned parameters be shared across tasks?* The landscape of existing deep MTL approaches can be organized based on how they answer this question at the joint network architecture level (Figure 1).

**Classical approaches.** Neural network MTL was first introduced in the case of shallow networks (Caruana, 1998), before deep networks were prevalent. The key idea was to add output neurons to predict auxiliary labels for related tasks, which would act as regularizers for the hidden representation. Many deep learning extensions remain close in nature to this approach, learning a shared representation at a high-level layer, followed by task-specific (i.e., unshared) decoders that extract labels for each task (Devin et al., 2016; Dong et al., 2015; Huang et al., 2013; 2015; Jaderberg et al., 2017; Liu et al., 2015a; Ranjan et al., 2016; Wu et al., 2015; Zhang et al., 2014) (Figure 1a). This approach can be extended to task-specific input encoders (Devin et al., 2016; Luong et al., 2016), and the underlying single-task model may be adapted to ease task integration (Ranjan et al., 2016; Wu et al., 2015), but the core network is still shared in its entirety.

**Column-based approaches.** Column-based approaches (Jou and Chang, 2016; Misra et al., 2016; Rusu et al., 2016; Yang and Hospedales, 2017), assign each task its own layer of task-specific parameters at each shared depth (Figure 1b). They then define a mechanism for sharing parameters between tasks at each shared depth, e.g., by having a shared tensor factor across tasks (Yang and Hospedales, 2017), or allowing some form of communication between columns (Jou and Chang,

2016; Misra et al., 2016; Rusu et al., 2016). Observations of negative effects of sharing in column-based methods (Rusu et al., 2016) can be attributed to mismatches between the features required at the same depth between tasks that are too dissimilar.

**Supervision at custom depths.** There may be an intuitive hierarchy describing how a set of tasks are related. Several approaches integrate supervised feedback from each task at levels consistent with such a hierarchy (Hashimoto et al., 2016; Toshniwal et al., 2017; Zhang and Weiss, 2016) (Figure 1c). This method can be sensitive to the design of the hierarchy (Toshniwal et al., 2017), and to which tasks are included therein (Hashimoto et al., 2016). One approach learns a task-relationship hierarchy during training (Lu et al., 2017), though learned parameters are still only shared across matching depths. Supervision at custom depths has also been extended to include explicit recurrence that reintegrates information from earlier predictions (Bilen and Vedaldi, 2016; Zamir et al., 2016). Although these recurrent methods still rely on pre-defined hierarchical relationships between tasks, they provide evidence of the potential of learning transformations that have a different function for different tasks at different depths, i.e., in this case, at different depths unrolled in time.

**Universal representations.** One approach shares all core model parameters except batch normalization scaling factors (Bilen and Vedaldi, 2017) (Figure 1d). When the number of classes is equal across tasks, even output layers can be shared, and the small number of task-specific parameters enables strong performance to be maintained. This method was applied to a diverse array of vision tasks, demonstrating the power of a small number of scaling parameters in adapting layer functionality for different tasks. This observation helps to motivate the method developed in Section 3.

## 2.2 THE PARALLEL ORDERING ASSUMPTION

A common interpretation of deep learning is that layers extract progressively higher level features at later depths (Lecun et al., 2015). A natural assumption is then that the learned transformations that extract these features are also tied to the depth at which they are learned. The core assumption motivating MTL is that regularities across tasks will result in learned transformations that can be leveraged to improve generalization. However, the methods reviewed in Section 2.1 add the further assumption that *subsequences of the feature hierarchy align across tasks and sharing between tasks occurs only at aligned depths* (Figure 1); we call this the *parallel ordering assumption*.

Consider $T$ tasks $t_1, \ldots, t_T$ to be learned jointly, with each $t_i$ associated with a model $y_i = \mathcal{F}_i(x_i)$. Suppose sharing across tasks occurs at $D$ consecutive depths. Let $\mathcal{E}_i$ ($\mathcal{D}_i$) be $t_i$'s task-specific encoder (decoder) to (from) the core sharable portion of the network from its inputs (to its outputs). Let $W_k^i$ be the layer of learned weights (e.g., affine or convolutional) for task $i$ at shared depth $k$, with $\phi_k$ an optional nonlinearity. The parallel ordering assumption implies

$$y_i = (\mathcal{D}_i \circ \phi_D \circ W_D^i \circ \phi_{D-1} \circ W_{D-1}^i \circ \ldots \circ \phi_1 \circ W_1^i \circ \mathcal{E}_i)(x_i), \text{ with } W_k^i \approx W_k^j \; \forall \, (i, j, k). \quad (1)$$

The approximate equality "$\approx$" means that at each shared depth the applied weight tensors for each task are similar and compatible for sharing. For example, learned parameters may be shared across all $W_k^i$ for a given $k$, but not between $W_k^i$ and $W_l^j$ for any $k \neq l$. For closely-related tasks, this assumption may be a reasonable constraint. However, as more tasks are added to a joint model, it may be more difficult for each layer to represent features of its given depth for all tasks. Furthermore, for very distant tasks, it may be unreasonable to expect that task feature hierarchies match up at all, even if the tasks are related intuitively. The conjecture explored in this paper is that parallel ordering limits the potential of deep MTL by the strong constraint it enforces on the use of each layer.

## 3 DEEP MULTITASK LEARNING WITH SOFT ORDERING OF LAYERS

Now that parallel ordering has been identified as a constricting feature of deep MTL approaches, its necessity can be tested, and the resulting observations can be used to develop more flexible methods.

### 3.1 A FOIL FOR THE PARALLEL ORDERING ASSUMPTION: PERMUTING SHARED LAYERS

Consider the most common deep MTL setting: hard-sharing of layers, where each layer in $\{W_k\}_{k=1}^D$ is shared in its entirety across all tasks. The baseline deep MTL model for each task $t_i$ is given by

$$y_i = (\mathcal{D}_i \circ \phi_D \circ W_D \circ \phi_{D-1} \circ W_{D-1} \circ \ldots \circ \phi_1 \circ W_1 \circ \mathcal{E}_i)(x_i). \quad (2)$$

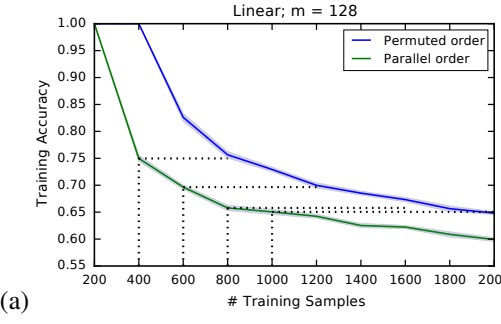 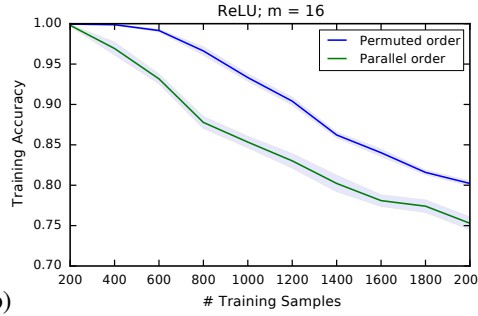

Figure 2: **Fitting two random tasks.** (a) The dotted lines show that permuted ordering fits $n$ samples as well as parallel fits $n/2$ for linear networks; (b) For ReLU networks, permuted ordering enjoys a similar advantage. Thus, permuted ordering of shared layers eases integration of information across disparate tasks.

This setup satisfies the parallel ordering assumption. Consider now an alternative scheme, equivalent to the above, except with learned layers applied in different orders for different task. That is,

$$y_i = (\mathcal{D}_i \circ \phi_D \circ W_{\rho_i(D)} \circ \phi_{D-1} \circ W_{\rho_i(D-1)} \circ \ldots \circ \phi_1 \circ W_{\rho_i(1)} \circ \mathcal{E}_i)(x_i), \tag{3}$$

where $\rho_i$ is a task-specific permutation of size $D$, and $\rho_i$ is fixed before training. If there are sets of tasks for which joint training of the model defined by Eq. 3 achieves similar or improved performance over Eq. 2, then parallel ordering is not a necessary requirement for deep MTL. Of course, in this formulation, it is required that the $W_k$ can be applied in any order. See Section 6 for examples of possible generalizations.

Note that this multitask permuted ordering differs from an approach of training layers in multiple orders for a single task. The single-task case results in a model with increased commutativity between layers, a behavior that has also been observed in residual networks (Veit et al., 2016), whereas here the result is *a set of layers that are assembled in different ways for different tasks*.

### 3.2 THE INCREASED EXPRESSIVITY OF PERMUTED ORDERING

**Fitting tasks of random patterns.** Permuted ordering is evaluated by comparing it to parallel ordering on a set of tasks. Randomly generated tasks (similar to (Kirkpatrick et al., 2017)) are the most disparate possible tasks, in that they share minimal information, and thus help build intuition for how permuting layers could help integrate information in broad settings. The following experiments investigate how accurately a model can jointly fit two tasks of $n$ samples. The data set for task $t_i$ is $\{(x_{ij}, y_{ij})\}_{j=1}^n$, with each $x_{ij}$ drawn uniformly from $[0, 1]^m$, and each $y_{ij}$ drawn uniformly from $\{0, 1\}$. There are two shared learned affine layers $W_k : \mathbb{R}^m \to \mathbb{R}^m$. The models with permuted ordering (Eq. 3) are given by

$$y_1 = (O \circ \phi \circ W_2 \circ \phi \circ W_1)(x_1) \text{ and } y_2 = (O \circ \phi \circ W_1 \circ \phi \circ W_2)(x_2), \tag{4}$$

where $O$ is a final shared classification layer. The reference parallel ordering models are defined identically, but with $W_k$ in the same order for both tasks. Note that fitting the parallel model with $n$ samples is equivalent to a single-task model with $2n$. In the first experiment, $m = 128$ and $\phi = I$. Although adding depth does not add expressivity in the single-task linear case, it is useful for examining the effects of permuted ordering, and deep linear networks are known to share properties with nonlinear networks (Saxe et al., 2013). In the second experiment, $m = 16$ and $\phi = \text{ReLU}$.

The results are shown in Figure 2. Remarkably, in the linear case, permuted ordering of shared layers does not lose accuracy compared to the single-task case. A similar gap in performance is seen in the nonlinear case, indicating that this behavior extends to more powerful models. Thus, the learned permuted layers are able to successfully adapt to their different orderings in different tasks.

Looking at conditions that make this result possible can shed further light on this behavior. For instance, consider $T$ tasks $t_1, \ldots, t_T$, with input and output size both $m$, and optimal linear solutions $F_1, \ldots, F_T$, respectively. Let $F_1, \ldots, F_T$ be $m \times m$ matrices, and suppose there exist matrices

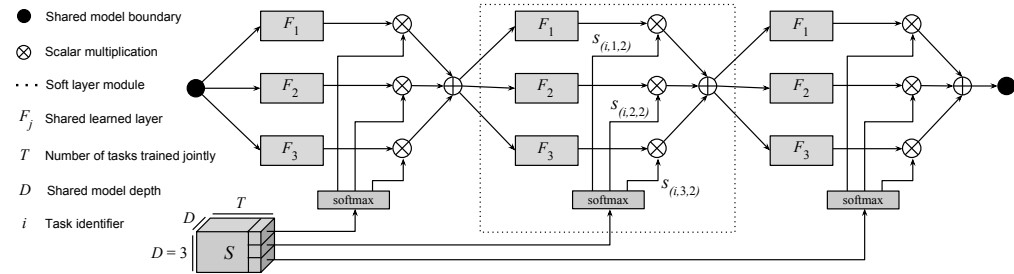

Figure 3: **Soft ordering of shared layers.** Sample soft ordering network with three shared layers. Soft ordering (Eq. 7) generalizes Eqs. 2 and 3, by learning a tensor $S$ of task-specific scaling parameters. $S$ is learned jointly with the $F_j$, to allow flexible sharing across tasks and depths. The $F_j$ in this figure each include a shared weight layer and any nonlinearity. This architecture enables the learning of layers that are used in different ways at different depths for different tasks.

$G_1, \ldots, G_T$ such that $F_i = G_i G_{(i+1 \bmod T)} \ldots G_{(i-1 \bmod T)} \ \forall \ i$. Then, because the matrix trace is invariant under cyclic permutations, the constraint arises that

$$\mathrm{tr}(F_1) = \mathrm{tr}(F_2) = \ldots = \mathrm{tr}(F_T). \tag{5}$$

In the case of random matrices induced by the random tasks above, the traces of the $F_i$ are all equal in expectation and concentrate well as their dimensionality increases. So, the restrictive effect of Eq. 5 on the expressivity of permuted ordering here is negligible.

**Adding a small number of task-specific scaling parameters.** Of course, real world tasks are generally much more structured than random ones, so such reliable expressivity of permuted ordering might not always be expected. However, adding a small number of task-specific scaling parameters can help adapt learned layers to particular tasks. This observation has been previously exploited in the parallel ordering setting, for learning task-specific batch normalization scaling parameters (Bilen and Vedaldi, 2017) and controlling communication between columns (Misra et al., 2016). Similarly, in the permuted ordering setting, the constraint induced by Eq. 5 can be reduced by adding task-specific scalars $\{s_i\}_{i=2}^T$ such that $F_i = s_i G_i G_{(i+1 \bmod T)} \ldots G_{(i-1 \bmod T)}$, and $s_1 = 1$. The constraint given by Eq. 5 then reduces to

$$\mathrm{tr}(F_i/s_i) = \mathrm{tr}(F_{i+1}/s_{i+1}) \ \forall \ 1 \leq i < T \implies s_{i+1} = s_i\big(\mathrm{tr}(F_{i+1})/\mathrm{tr}(F_i)\big), \tag{6}$$

which are defined when $\mathrm{tr}(F_i) \neq 0 \ \forall \ i < T$. Importantly, the number of task-specific parameters does not depend on $m$, which is useful for scalability as well as encouraging maximal sharing between tasks. The idea of using a small number of task-specific scaling parameters is incorporated in the soft ordering approach introduced in the next section.

### 3.3 SOFT ORDERING OF SHARED LAYERS

Permuted ordering tests the parallel ordering assumption, but still fixes an *a priori* layer ordering for each task before training. Here, a more flexible *soft ordering* approach is introduced, which allows jointly trained models to learn *how* layers are applied while simultaneously learning the layers themselves. Consider again a core network of depth $D$ with layers $W_1, \ldots, W_D$ learned and shared across tasks. The soft ordering model for task $t_i$ is defined as follows:

$$y_i^k = \sum_{j=1}^{D} s_{(i,j,k)}(\phi_k[W_j(y_i^{k-1})]), \text{ with } \sum_{j=1}^{D} s_{(i,j,k)} = 1 \ \forall \ (i,k), \tag{7}$$

where $y_i^0 = \mathcal{E}_i(x_i)$, $y_i = \mathcal{D}_i(y_i^D)$, and each $s_{(i,j,k)}$ is drawn from $S$: a tensor of learned scales for each task $t_i$ for each layer $W_j$ at each depth $k$. Figure 3 shows an example of a resulting depth three model. Motivated by Section 3.2 and previous work (Misra et al., 2016), $S$ adds only $D^2$ scaling parameters per task, which is notably not a function of the size of any $W_j$. The constraint that all $s_{(i,j,k)}$ sum to 1 for any $(i,k)$ is implemented via softmax, and emphasizes the idea that a soft *ordering* is what is being learned; in particular, this formulation subsumes any fixed layer ordering $\rho_i$ by $s_{(i,\rho_i(k),k)} = 1 \ \forall \ (i,k)$. $S$ can be learned jointly with the other learnable parameters

in the $W_k$, $\mathcal{E}_i$, and $\mathcal{D}_i$ via backpropagation. In training, all $s_{(i,j,k)}$ are initialized with equal values, to reduce initial bias of layer function across tasks. It is also helpful to apply dropout after each shared layer. Aside from its usual benefits (Srivastava et al., 2014), dropout has been shown to be useful in increasing the generalization capacity of shared representations (Devin et al., 2016). Since the trained layers in Eq. 7 are used for different tasks and in different locations, dropout makes them more robust to supporting different functionalities. These ideas are tested empirically on the MNIST, UCI, Omniglot, and CelebA data sets in the next section.

## 4    EMPIRICAL EVALUATION OF SOFT LAYER ORDERING

These experiments evaluate soft ordering against fixed ordering MTL and single-task learning. The first experiment applies them to intuitively related MNIST tasks, the second to "superficially un-related" UCI tasks, the third to the real-world problem of Omniglot character recognition, and the fourth to large-scale facial attribute recognition. In each experiment, single task, parallel ordering (Eq. 2), permuted ordering (Eq. 3), and soft ordering (Eq. 7) train an equivalent set of core layers. In permuted ordering, the order of layers were randomly generated for each task each trial. See Appendix A for additional details, including additional details specific to each experiment.

### 4.1    DISENTANGLING RELATED TASKS: MNIST DIGIT$_1$-VS.-DIGIT$_2$ BINARY CLASSIFICATION

This experiment evaluates the ability of multitask methods to exploit tasks that are intuitively related, but have disparate input representations. Binary classification problems derived from the MNIST hand-written digit dataset are a common test bed for evaluating deep learning methods that require multiple tasks (Fernando et al., 2017; Kirkpatrick et al., 2017; Yang and Hospedales, 2017). Here, the goal of each task is to distinguish between two distinct randomly selected digits. To create initial dissimilarity across tasks that multitask models must disentangle, each $\mathcal{E}_i$ is a random frozen fully-connected ReLU layer with output size 64. There are four core layers, each a fully-connected ReLU layer with 64 units. Each $\mathcal{D}_i$ is an unshared dense layer with a single sigmoid classification output.

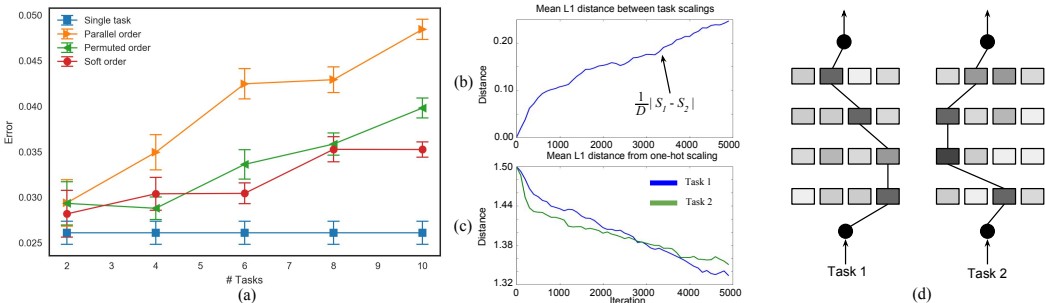

Figure 4: **MNIST results.** (a) Relative performance of permuted and soft ordering compared to parallel ordering improves as the number of tasks increases, showing how flexibility of order can help in scaling to more tasks. Note that cost savings of multitask over single task models in terms of number of trainable parameters scales linearly with the number of tasks. For a representative two-task soft order experiment (b) the layer-wise distance between scalings of the tasks increases by iteration, and (c) the scalings move towards a hard ordering. (d) The final learned relative scale of each shared layer at each depth for each task is indicated by shading, with the strongest path drawn, showing that a distinct soft order is learned for each task (• marks the shared model boundary).

Results are shown in Figure 4. The relative performance of permuted ordering and soft ordering compared to parallel ordering increases with the number of tasks trained jointly (Figure 4a), showing how flexibility of order can help in scaling to more tasks. This result is consistent with the hypothesis that parallel ordering has increased negative effects as the number of tasks increases. Figure 4b-d show what soft ordering actually learns: The scalings for tasks diverge as layers specialize to different functions for different tasks.

(a)

| Dataset | Input Features | Output classes | Samples |
|---|---|---|---|
| Australian credit | 14 | 2 | 690 |
| Breast cancer | 30 | 2 | 569 |
| Ecoli | 7 | 8 | 336 |
| German credit | 24 | 2 | 1000 |
| Heart disease | 13 | 5 | 303 |
| Hepatitis | 19 | 2 | 155 |
| Iris | 4 | 3 | 150 |
| Pima diabetes | 8 | 2 | 768 |
| Wine | 13 | 3 | 178 |
| Yeast | 8 | 10 | 1484 |

(b)

Figure 5: **UCI data sets and results.** (a) The ten UCI tasks used in joint training; the varying types of problems and dataset characteristics show the diversity of this set of tasks. (b) Mean test error over all ten tasks by iteration. Permuted and parallel order show no improvement after the first 1000 iterations, while soft order decisively outperforms the other methods.

## 4.2 SUPERIFICALLY UNRELATED TASKS: JOINT TRAINING OF TEN POPULAR UCI DATASETS

The next experiment evaluates the ability of soft ordering to integrate information across a diverse set of "superficially unrelated" tasks (Mahmud and Ray, 2008), i.e., tasks with no immediate intuition for how they may be related. Ten tasks are taken from some of most popular UCI classification data sets (Lichman, 2013). Descriptions of these tasks are given in Figure 5a. Inputs and outputs have no a priori shared meaning across tasks. Each $\mathcal{E}_i$ is a learned fully-connected ReLU layer with output size 32. There are four core layers, each a fully-connected ReLU layer with 32 units. Each $\mathcal{D}_i$ is an unshared dense softmax layer for the given number of classes. The results in Figure 5(b) show that, while parallel and permuted show no improvement in error after the first 1000 iterations, soft ordering significantly outperforms the other methods. With this flexible layer ordering, the model is eventually able to exploit significant regularities underlying these seemingly disparate domains.

## 4.3 EXTENSION TO CONVOLUTIONS: MULTI-ALPHABET CHARACTER RECOGNITION

The Omniglot dataset (Lake et al., 2015) consists of fifty alphabets, each of which induces a different character recognition task. Deep MTL approaches have recently shown promise on this dataset (Yang and Hospedales, 2017). It is a useful benchmark for MTL because the large number of tasks allows analysis of performance as a function of the number of tasks trained jointly, and there is clear intuition for how knowledge of some alphabets will increase the ability to learn others. Omniglot is also a good setting for evaluating the ability of soft ordering to learn how to compose layers in different ways for different tasks: it was developed as a problem with inherent composibility, e.g., similar kinds of strokes are applied in different ways to draw characters from different alphabets (Lake et al., 2015). Consequently, it has been used as a test bed for deep generative models (Rezende et al., 2016). To evaluate performance for a given number of tasks $T$, a single random ordering of tasks was created, from which the first $T$ tasks are considered. Train/test splits are created in the same way as previous work (Yang and Hospedales, 2017), using 10% or 20% of data for testing.

This experiment is a scale-up of the previous experiments in that it evaluates soft ordering of convolutional layers. The models are made as close as possible in architecture to previous work (Yang and Hospedales, 2017), while allowing soft ordering to be applied. There are four core layers, each convolutional followed by max pooling. $\mathcal{E}_i(x_i) = x_i \; \forall \; i$, and each $\mathcal{D}_i$ is a fully-connected softmax layer with output size equal to the number of classes. The results show that soft ordering is able to consistently outperform other deep MTL approaches (Figure 6). The improvements are robust to the number of tasks (Figure 6a) and the amount of training data (Figure 6c), suggesting that soft ordering, not task complexity or model complexity, is responsible for the improvement.

Permuted ordering performs significantly worse than parallel ordering in this domain. This is not surprising, as deep vision systems are known to induce a common feature hierarchy, especially within the first couple of layers (Lee et al., 2008; Lecun et al., 2015). Parallel ordering has this hierarchy built in; for permuted ordering it is more difficult to exploit. However, the existence of this feature hierarchy does not preclude the possibility that the functions (i.e., layers) used to produce the hierarchy may be useful in other contexts. Soft ordering allows the discovery of such uses. Figure 6b shows how each layer is used more or less at different depths. The soft ordering model learns a "soft

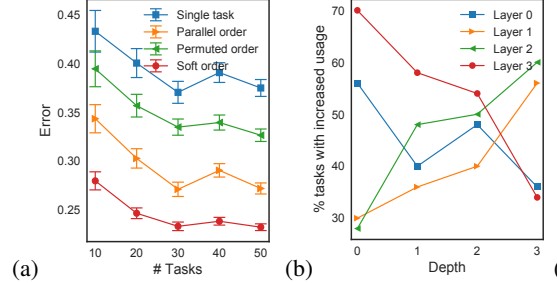

Figure 6: **Omniglot results.** (a) Error by number of tasks trained jointly. Soft ordering significantly outperforms single task and both fixed ordering approaches for each number of tasks; (b) Distribution of learned layer usage by depth across all 50 tasks for a soft order run. The usage of each layer is correlated (or inversely correlated) with depth. This coincides with the understanding that there is some innate hierarchy in convolutional networks, which soft ordering is able to discover. For instance, the usage of Layer 3 decreases as the depth increases, suggesting that its primary purpose is low-level feature extraction, though it is still sees substantial use in deeper contexts; (c) Errors with all 50 tasks for different training set sizes. The first five methods are previous deep MTL results (Yang and Hospedales, 2017), which use multitask tensor factorization methods in a shared parallel ordering. Soft ordering significantly outperforms the other approaches, showing the approach scales to real-world tasks requiring specialized components such as convolutional layers.

hierarchy" of layers, in which each layer has a distribution of increased or decreased usage at each depth. In this case, the usage of each layer is correlated (or inversely correlated) with depth. For instance, the usage of Layer 3 decreases as the depth increases, suggesting that its primary purpose is low-level feature extraction, though it is still sees substantial use in deeper contexts. Section 5 describes an experiment that further investigates the behavior of a single layer in different contexts.

## 4.4 LARGE-SCALE APPLICATION: FACIAL ATTRIBUTE RECOGNITION

Although facial attributes are all high-level concepts, they do not intuitively exist at the same level of a shared hierarchy (even one that is learned; Lu et al., 2017). Rather, these concepts are related in multiple subtle and overlapping ways in semantic space. This experiment investigates how a soft ordering approach, as a component in a larger system, can exploit these relationships.

The CelebA dataset consists of $\approx$200K $178 \times 218$ color images, each with binary labels for 40 facial attributes (Liu et al., 2015b). In this experiment, each label defines a task, and parallel and soft order models are based on a ResNet-50 vision model (He et al., 2016), which has also been used in recent state-of-the-art approaches to CelebA (Günther et al., 2017; He et al., 2017). Let $\mathcal{E}_i$ be a ResNet-50 model truncated to the final average pooling layer, followed by a linear layer projecting the embedding to size 256. $\mathcal{E}_i$ is shared across all tasks. There are four core layers, each a dense ReLU layer with 256 units. Each $\mathcal{D}_i$ is an unshared dense sigmoid layer. Parallel ordering and soft ordering models were compared. To further test the robustness of learning, models were trained with and without the inclusion of an additional facial landmark detection regression task. Soft order models were also tested with and without the inclusion of a fixed identity layer at each depth. The identity layer can increase consistency of representation across contexts, which can ease learning of each layer, while also allowing soft ordering to tune how much total non-identity transformation to use for each individual task. This is especially relevant for the case of attributes, since different tasks can have different levels of complexity and abstraction.

The results are given in Figure 7c. Existing work that used a ResNet-50 vision model showed that using a parallel order multitask model improved test error over single-task learning from 10.37 to 9.58 (He et al., 2017). With our faster training strategy and the added core layers, our parallel ordering model achieves a test error of 10.21. The soft ordering model yielded a substantial improvement beyond this to 8.79, demonstrating that soft ordering can add value to a larger deep learning system. Including landmark detection yielded a marginal improvement to 8.75, while for parallel ordering it degraded performance slightly, indicating that soft ordering is more robust to joint training of diverse kinds of tasks. Including the identity layer improved performance to 8.64, though with both the land-

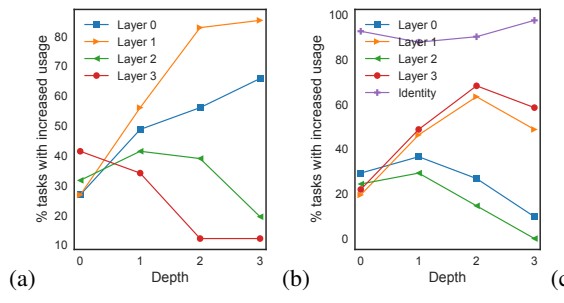

| Deep MTL method | Test Error % |
|---|---|
| Single Task (He et al., 2017) | 10.37 |
| MTL Baseline (He et al., 2017) | 9.58 |
| Parallel Order | 10.21 |
| Parallel Order + Landmarks | 10.29 |
| Soft Order | 8.79 |
| Soft Order + Landmarks | 8.75 |
| Soft Order + Identity | **8.64** |
| Soft Order + Landmarks + Identity | 8.68 |

Figure 7: **CelebA results.** Layer usage by depth (a) without and (b) with inclusion of the identity layer. In both cases, layers with lower usage at lower depths have higher usage at higher depths, and vice versa. The identity layer almost always sees increased usage; its application can increase consistency of representation across contexts. (c) Soft order models achieve a significant improvement over parallel ordering, and receive a boost from including the identity layer. The first two rows are previous work with ResNet-50 that show their baseline improvement from single task to multitask.

mark detection and the identity layer this improvement was slightly diminished. One explanation for this degradation is that the added flexibility provided by the identity layer offsets the regularization provided by landmark detection. Note that previous work has shown that adaptive weighting of task loss (He et al., 2017; Rudd et al., 2016), data augmentation and ensembling (Günther et al., 2017), and a larger underlying vision model (Lu et al., 2017) each can also yield significant improvements. Aside from soft ordering, none of these improvements alter the *multitask topology*, so their benefits are expected to be complementary to that of soft ordering demonstrated in this experiment. By coupling them with soft ordering, greater improvements should be possible.

Figures 7a-b characterize the usage of each layer learned by soft order models. Like in the case of Omniglot, layers that are used less at lower depths are used more at higher depths, and vice versa, giving further evidence that the models learn a "soft hierarchy" of layer usage. When the identity layer is included, its usage is almost always increased through training, as it allows the model to use smaller specialized proportions of nonlinear structure for each individual task.

## 5 VISUALIZING THE BEHAVIOR OF SOFT ORDERING LAYERS

The success of soft layer ordering suggests that layers learn functional primitives with similar effects in different contexts. To explore this idea qualitatively, the following experiment uses generative visual tasks. The goal of each task is to learn a function $(x, y) \rightarrow v$, where $(x, y)$ is a pixel coordinate and $v$ is a brightness value, all normalized to $[0, 1]$. Each task is defined by a single image of a "4" drawn from the MNIST dataset; all of its pixels are used as training data. Ten tasks are trained using soft ordering with four shared dense ReLU layers of 100 units each. $\mathcal{E}_i$ is a linear encoder that is shared across tasks, and $\mathcal{D}_i$ is a global average pooling decoder. Thus, task models are distinguished completely by their learned soft ordering scaling parameters $s_t$. To visualize the behavior of layer $l$ at depth $d$ for task $t$, the predicted image for task $t$ is generated across varying magnitudes of $s_{(t,l,d)}$. The results for the first two tasks and the first layer are shown in Table 1. Similar function is observed in each of the six contexts, suggesting that the layers indeed learn functional primitives.

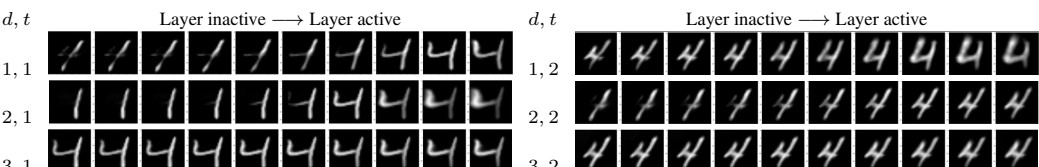

Table 1: **Example behavior of a soft order layer**. For each task $t$, and at each depth $d$, the effect of increasing the activation of of this particular layer is to expand the left side of the "4" in a manner appropriate to the functional context (e.g., the magnitude of the effect decreases with depth). Results for other layers are similar, suggesting that the layers implement functional primitives.

## 6 DISCUSSION AND FUTURE WORK

In the interest of clarity, the soft ordering approach in this paper was developed as a relatively small step away from the parallel ordering assumption. To develop more practical and specialized methods, inspiration can be taken from recurrent architectures, the approach can be extended to layers of more general structure, and applied to training and understanding general functional building blocks.

**Connections to recurrent architectures.** Eq. 7 is defined recursively with respect to the learned layers shared across tasks. Thus, the soft-ordering architecture can be viewed as a new type of recurrent architecture designed specifically for MTL. From this perspective, Figure 3 shows an unrolling of a *soft layer module*: different scaling parameters are applied at different depths when unrolled for different tasks. Since the type of recurrence induced by soft ordering does not require task input or output to be sequential, methods that use recurrence in such a setting are of particular interest (Liang and Hu, 2015; Liao and Poggio, 2016; Pinheiro and Collobert, 2014; Socher et al., 2011; Zamir et al., 2016). Recurrent methods can also be used to reduce the size of $S$ below $O(TD^2)$, e.g., via recurrent hypernetworks (Ha et al., 2016). Finally, Section 4 demonstrated soft ordering where shared learned layers were fully-connected or convolutional; it is also straightforward to extend soft ordering to shared layers with internal recurrence, such as LSTMs (Hochreiter and Schmidhuber, 1997). In this setting, soft ordering can be viewed as inducing a higher-level recurrence.

**Generalizing the structure of shared layers.** For clarity, in this paper all core layers in a given setup had the same shape. Of course, it would be useful to have a generalization of soft ordering that could subsume any modern deep architecture with many layers of varying structure. As given by Eq. 7, soft ordering requires the same shape inputs to the element-wise sum at each depth. Reshapes and/or resampling can be added as adapters between tensors of different shape; alternatively, a function other than a sum could be used. For example, instead of learning a weighting across layers at each depth, a probability of applying each module could be learned in a manner similar to adaptive dropout (Ba and Frey, 2013; Li et al., 2016) or a sparsely-gated mixture of experts (Shazeer et al., 2017). Furthermore, the idea of a soft ordering of layers can be extended to soft ordering over modules with more general structure, which may more succinctly capture recurring modularity.

**Training generalizable building blocks.** Because they are used in different ways at different locations for different tasks, the shared trained layers in permuted and soft ordering have learned more general functionality than layers trained in a fixed location or for a single task. A natural hypothesis is that they are then more likely to generalize to future unseen tasks, perhaps even without further training. This ability would be especially useful in the small data regime, where the number of trainable parameters should be limited. For example, given a collection of these layers trained on a previous set of tasks, a model for a new task could learn how to apply these building blocks, e.g., by learning a soft order, while keeping their internal parameters fixed. Learning an efficient set of such generalizable layers would then be akin to learning a set of *functional primitives*. Such functional modularity and repetition is evident in the natural, technological and sociological worlds, so such a set of functional primitives may align well with complex real-world models. This perspective is related to recent work in reusing modules in the parallel ordering setting (Fernando et al., 2017). The different ways in which different tasks learn to use the same set of modules can also help shed light on how tasks are related, especially those that seem superficially disparate (e.g., by extending the analysis performed for Figure 4d), thus assisting in the discovery of real-world regularities.

## 7 CONCLUSION

This paper has identified *parallel ordering* of shared layers as a common assumption underlying existing deep MTL approaches. This assumption restricts the kinds of shared structure that can be learned between tasks. Experiments demonstrate how direct approaches to removing this assumption can ease the integration of information across plentiful and diverse tasks. *Soft ordering* is introduced as a method for learning how to apply layers in different ways at different depths for different tasks, while simultaneously learning the layers themselves. Soft ordering is shown to outperform parallel ordering methods as well as single-task learning across a suite of domains. These results show that deep MTL can be improved while generating a compact set of multipurpose functional primitives, thus aligning more closely with our understanding of complex real-world processes.

ACKNOWLEDGMENTS

We would like to thank Matt Feiszli for valuable discussions and all anonymous reviewers for their helpful feedback.

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

# A  EXPERIMENTAL DETAILS

All experiments were run with the Keras deep learning framework Chollet et al. (2015), using the Tensorflow backend (Abadi et al., 2015). All experiments used the Adam optimizer with default parameters (Kingma and Ba, 2014) unless otherwise specified.

In each iteration of multitask training, a random batch for each task is processed, and the results are combined across tasks into a single update. Compared to alternating batches between tasks (Luong et al., 2016), processing all tasks simultaneously simplified the training procedure, and led to faster and lower final convergence. When encoders are shared, the inputs of the samples in each batch are the same across tasks. Cross-entropy loss was used for all classification tasks. The overall validation loss is the sum over all per task validation losses.

In each experiment, single task, parallel ordering (Eq. 2), permuted ordering (Eq. 3), and soft ordering (Eq. 7) trained an equivalent set of core layers. In permuted ordering, the order of layers was randomly generated for each task each trial. Several trials were run for each setup to produce confidence bounds.

## A.1  MNIST EXPERIMENTS

Input pixel values were normalized to be between 0 and 1. The training and test sets for each task were the MNIST train and test sets restricted to the two selected digits. A dropout rate of 0.5 was applied at the output of each core layer. Each setup was trained for 20K iterations, with each batch consisting of 64 samples for each task.

When randomly selecting the pairs of digits that define a set of tasks, digits were selected without replacement within a task, and with replacement across tasks, so there were 45 possible tasks, and $45^k$ possible sets of tasks of size $k$.

## A.2  UCI EXPERIMENTS

For all tasks, each input feature was scaled to be between 0 and 1. For each task, training and validation data were created by a random 80-20 split. This split was fixed across trials. A dropout rate of 0.8 was applied at the output of each core layer.

## A.3  OMNIGLOT EXPERIMENTS

To enable soft ordering, the output of all shared layers must have the same shape. For comparability, the models were made as close as possible in architecture to previous work (Yang and Hospedales, 2017), in which models had four sharable layers, three of which were 2D convolutions followed by $2 \times 2$ max-pooling, of which two had $3 \times 3$ kernels. So, in this experiment, to evaluate soft

ordering of convolutional layers, there were four core layers, each a 2D convolutional layer with ReLU activation and kernel size $3 \times 3$. Each convolutional layer was followed by a $2 \times 2$ max-pooling layer. The number of filters for each convolutional layer was set at 53, which makes the number of total model parameters as close as possible to the reference model. A dropout rate of 0.5 was applied at the output of after each core layer.

The Omniglot dataset consists of $105 \times 105$ black-and-white images. There are fifty alphabets of characters and twenty images per character. To be compatible with the shapes of shared layers, the input was zero-padded along the third dimension so that its shape was $105 \times 105 \times 53$, i.e., with the first $105 \times 105$ slice containing the image data and the remainder zeros. To evaluate approaches on $k$ tasks, a random ordering of the fifty tasks was created and fixed across all trials. In each trial, the first $k$ tasks in this ordering were trained jointly for 5000 iterations, with each training batch containing $k$ random samples, one from each task. The fixed ordering of tasks was as follows:

[Gujarati, Sylheti, Arcadian, Tibetan, Old Church Slavonic (Cyrillic), Angelic, Malay (Jawi-Arabic), Sanskrit, Cyrillic, Anglo-Saxon Futhorc, Syriac (Estrangelo), Ge'ez, Japanese (katakana), Keble, Manipuri, Alphabet of the Magi, Gurmukhi, Korean, Early Aramaic, Atemayar Qelisayer, Tagalog, Mkhedruli (Georgian), Inuktitut (Canadian Aboriginal Syllabics), Tengwar, Hebrew, N'Ko, Grantha, Latin, Syriac (Serto), Tifinagh, Balinese, Mongolian, ULOG, Futurama, Malayalam, Oriya, Ojibwe (Canadian Aboriginal Syllabics), Avesta, Kannada, Bengali, Japanese (hiragana), Armenian, Aurek-Besh, Glagolitic, Asomtavruli (Georgian), Greek, Braille, Burmese (Myanmar), Blackfoot (Canadian Aboriginal Syllabics), Atlantean].

## A.4 CELEBA EXPERIMENTS

The training, validation, and test splits provided by Liu et al. (2015b) were used. There are $\approx$160K images for training, $\approx$20K for validation, and $\approx$20K for testing. The dataset contains 20 images of each of approximately $\approx$10K celebrities. The images for a given celebrity occur in only one of the three dataset splits, so models must also generalize to new human identities.

The weights for ResNet-50 were initialized with the pre-trained imagenet weights provided in the Keras framework Chollet et al. (2015). Image preprocessing was done with the default Keras image preprocessing function, including resizing all images to $224 \times 224$.

The output for the facial landmark detection task is a 10 dimensional vector indicating the $(x, y)$ locations of five landmarks, normalized between 0 and 1. Mean squared error was used as the training loss. When landmark detection is included, the target metric is still attribute classification error. This is because the aligned CelebA images are used, so accurate landmark detection is not a challenge, but including it as an additional task can still provide additional regularization to a multitask model.

A dropout rate of 0.5 was applied at the output of after each core layer. The experiments used a batch size of 32. After validation loss converges via Adam, models are trained with RMSProp with learning rate $1e^{-5}$, which is a similar approach to that used by Günther et al. (2017).

## A.5 EXPERIMENTS ON VISUALIZING LAYER BEHAVIOR

To produce the resulting image for a fixed model, the predictions at each pixel locations were generated, denormalized, and mapped back to the pixel coordinate space. The loss used for this experiment was mean squared error (MSE). Since all pixels for a task image are used for training, there is no sense of generalization to unseen data within a task. As a result, no dropout was used in this experiment.

Task models are distinguished completely by their learned soft ordering scaling parameters $s_t$, so the joint model can be viewed as a generative model which generates different 4's for varying values of $s_t$. To visualize the behavior of layer $l$ at depth $d$ for task $t$, the output of the model for task $t$ was visualized while sweeping $s_{(t,l,d)}$ across $[0, 1]$. To enable this sweeping while keeping the rest of the model behavior fixed, the softmax for each task at each depth was replaced with a sigmoid activation. Note that due to the global avgerage pooling decoder, altering the weight of a single layer has no observable effect at depth four.

