# OpenReview forum: "Beyond Shared Hierarchies: Deep Multitask Learning through Soft Layer Ordering"
_ICLR.cc/2018/Conference — Accept (Poster)_

### Official Review · AnonReviewer1 · 2017-11-26
**Interesting approach but lacks interpretation**

**Rating:** 6
**Confidence:** 3

**Review:**

This paper proposes a new approach for multi-task learning. While previous approaches assumes the order of shared layers are the same between tasks, this paper assume the order can vary across tasks, and the (soft) order is learned during training.  They show improved performance on a number of multi-task learning problems.

My primary concern about this paper is the lack of interpretation on permuting the layers. For example, in standard vision systems, low level filters "V1" learn edge detectors (gabor filters) and higher level filters learn angle detectors [1]. It is confusing why permuting these filters make sense. They accept different inputs (raw pixels vs edges). Moreover, if the network contains pooling layers, different locations of the pooling layer result in different shapes of the feature map, and the soft ordering strategy Eq. (7) does not work.

It makes sense that the more flexible model proposed by this paper performs better than previous models. The good aspect of this paper is that it has some performance improvements. But I still wonder the effect of permuting the layers. The paper also needs more clarifications in the writing. For example, in Section 3.3, how each s_(i, j, k) is sampled from S? The "parallel ordering" terminology also seems to be arbitrary...

[1] Lee, Honglak, Chaitanya Ekanadham, and Andrew Y. Ng. "Sparse deep belief net model for visual area V2." Advances in neural information processing systems. 2008.

---

> ### Author Response · Authors · 2018-01-03
> **Response to AnonReviewer1**
>
> Reviewer Comment: "My primary concern about this paper is the lack of interpretation on permuting the layers. For example, in standard vision systems, low level filters "V1" learn edge detectors (gabor filters) and higher level filters learn angle detectors [1]. It is confusing why permuting these filters make sense. They accept different inputs (raw pixels vs edges)."
>
> Response:
> We have added a new analysis to clarify this effect in the Omniglot and CelebA experiments. The main takeaway is that there may indeed be some amount of useful shared feature hierarchy across tasks, but the functions (i.e., layers) used to produce this hierarchy may also be useful in other contexts. Soft ordering allows this hierarchy to be exploited, and these additional uses to be discovered. This hierarchy is especially salient in the case of convolutional layers, which explains why parallel ordering does better than permuted ordering in the Omniglot experiments. For more details, see Section 4.3: paragraph 3 and Figure 6a and b; and Section 4.4: paragraph 4 and Figure 7a and 7b. These sections and figures are new, i.e., they’ve been added to the paper in response to this suggestion.
>
> Reviewer Comment: "Moreover, if the network contains pooling layers, different locations of the pooling layer result in different shapes of the feature map, and the soft ordering strategy Eq. (7) does not work."
>
> Response: In the Omniglot experiments, pooling layers are included, and Eq. (7) does work, because the pool size, kernel size and number of filters is the same for each layer. In this setting, through soft ordering, the same layers are effectively applied at different resolutions or scales. Extending Eq. (7) to the case of layers that produce feature maps of conflicting shapes is left to future work (Section 6).
>
> Reviewer Comment: "The paper also needs more clarifications in the writing. For example, in Section 3.3, how each s_(i, j, k) is sampled from S?"
>
> Reponse: s_(i, j, k) is retrieved from S via tensor indexing. The commas and parentheses are included for disambiguation, e.g., in the case where j = \rho_i(k).
>
> Reviewer Comment: "The "parallel ordering" terminology also seems to be arbitrary..."
>
> Response: The term "parallel ordering" is intended to capture the commonalities shared by the methods reviewed in Section 2.1: the sets of sharable layers available at two distinct depths do not intersect. In that sense it isn’t arbitrary but chosen specifically to establish contrast with the other approaches.

---

> > ### Comment · AnonReviewer1 · 2018-01-06
> > **The clarifications makes sense**
> >
> > Thanks for the additional experiments. The arguments and results make sense. Maybe soft ordering works for visual tasks because
> > 1) It is like the inception architecture, where the model can choose different filters by itself on each layer,
> > 2) It is like an RNN.
> > I am convinced that soft ordering should work well.
> >
> > The assumption on the same shape across all layers is still limited though.

---

### Official Review · AnonReviewer3 · 2017-11-27
**Deep MTL through Soft Layer Ordering Review**

**Rating:** 7
**Confidence:** 4

**Review:**

Summary: This paper proposes a different approach to deep multi-task learning using “soft ordering.”  Multi-task learning encourages the sharing of learned representations across tasks, thus using less parameters and tasks help transfer useful knowledge across. Thus enabling the reuse of universally learned representations and reuse them by assembling them in novel ways for new unseen tasks. The idea of “soft ordering” enforces the idea that there shall not be a rigid structure for all the tasks, but a soft structure would make the models more generalizable and modular.

The methods reviewed prior work which the authors refer to as “parallel order”, which assumed that subsequences of the feature hierarchy align across tasks and sharing between tasks occurs only at aligned depths whereas in this work the authors argue that this shouldn’t be the case. They authors then extend the approach to “permuted order” and finally present their proposed “soft ordering” approach. The authors argue that their proposed soft ordering approach increase the expressivity of the model while preserving the performance.

The “soft ordering” approach simply enable task specific selection of layers, scaled with a learned scaling factor, to be combined in which order to result for the best performance for each task. The authors evaluate their approach on MNIST, UCI, Omniglot and CelebA datasets and compare their approach to “parallel ordering” and “permuted ordering” and show the performance gain.

Positives:
- The paper is clearly written and easy to follow
- The idea is novel and impactful if its evaluated properly and consistently
- The authors did a great job summarizing prior work and motivating their approach

Negatives:
- Multi-class classification problem is one incarnation of Multi-Task Learning, there are other problems where the tasks are different (classification and localization) or auxiliary (depth detection for navigation). CelebA dataset could have been a good platform for testing different tasks, attribute classification and landmark detection.
(TODO) I would recommend that the authors test their approach on such setting.
- Figure 6 is a bit confusing, the authors do not explain why the “Permuted Order” performs worse than “Parallel Order”. Their assumptions and results as of this section should be consistent that soft order>permuted order>parallel order>single task.
 (TODO) I would suggest that the authors follow up on this result, which would be beneficial for the reader.
- Figure 4(a) and 5(b), the results shown on validation loss, how about testing error similar to Figure 6(a)? How about results for CelebA dataset, it could be useful to visualize them as was done for MNIST, Omniglot and UCL.
(TODO) I would suggest that the authors make the results consistent across all datasets and use the same metric such that its easy to compare.

Notation and Typos:
- Figure 2 is a bit confusing, how come the accuracy decreases with increasing number of training samples? Please clarify.
1- If I assume that the Y-Axis is incorrectly labeled and it is Training Error instead, then the permuted order is doing worse than the parallel order.
 2- If I assume that the X-Axis is incorrectly labeled and the numbering is reversed (start from max and ending at 0), then I think it would make sense.
- Figure 4 is very small and not easy to read the text. Does single task mean average performance over the tasks?
- In eq.(3) Choosing \sigma_i for a task-specific permutation of the network is a bit confusing, since it could be thought of as a sigmoid function, I suggest using a different symbol.
 Conclusion: I would suggest that the authors address the concerns mentioned above. Their approach and idea is very interesting and relevant, and addressing these suggestions will make the paper strong for publication.

---

> ### Author Response · Authors · 2018-01-03
> **Response to AnonReviewer3**
>
> Reviewer Comment: "Multi-class classification problem is one incarnation of Multi-Task Learning, there are other problems where the tasks are different (classification and localization) or auxiliary (depth detection for navigation). CelebA dataset could have been a good platform for testing different tasks, attribute classification and landmark detection.
> (TODO) I would recommend that the authors test their approach on such setting."
>
> Response: As suggested, we have added setups that include landmark detection as an additional task in Section 4.4. This additional task yielded a marginal improvement in performance for soft ordering, while it yielded a degredation for parallel ordering, showing that soft ordering can more easily handle such different kinds of tasks.
>
> Reviewer Comment: "Figure 6 is a bit confusing, the authors do not explain why the “Permuted Order” performs worse than “Parallel Order”. Their assumptions and results as of this section should be consistent that soft order>permuted order>parallel order>single task.
>  (TODO) I would suggest that the authors follow up on this result, which would be beneficial for the reader."
>
> Response: We have added a new analysis to clarify this effect in the Omniglot and CelebA experiments. The main takeaway is that there may indeed be some amount of useful shared hierarchy across tasks, but the functions (i.e., layers) used to produce this hierarchy may also be useful in other contexts. Soft ordering allows this hierarchy to be exploited, and these additional uses to be discovered. This hierarchy is especially salient in the case of convolutional layers, which explains why parallel ordering does better than permuted ordering in the Omniglot experiments. For more details, see Section 4.3: paragraph 3 and Figure 6a and b; and Section 4.4: paragraph 4 and Figure 7a and 7b. These sections and figures are new, i.e., they’ve been added to the paper in response to this suggestion.
>
> Reviewer Comment: "Figure 4(a) and 5(b), the results shown on validation loss, how about testing error similar to Figure 6(a)? How about results for CelebA dataset, it could be useful to visualize them as was done for MNIST, Omniglot and UCL.
> (TODO) I would suggest that the authors make the results consistent across all datasets and use the same metric such that its easy to compare."
>
> Response: As suggested, we have updated the figures to report test error, and have added a figure (Figure 7) to visualize the CelebA results.
>
> Reviewer Comment: "Figure 2 is a bit confusing, how come the accuracy decreases with increasing number of training samples? Please clarify."
>
> Response: Figure 2 reports the abilities of parallel and permuted ordering models to fit tasks of random data. This is a test of model expressivity, so accuracy is reported on the training set, and as the size of the training set increases it becomes more difficult for the models to memorize it.
>
> Reviewer Comment: "Figure 4 is very small and not easy to read the text. Does single task mean average performance over the tasks?"
>
> Response: We have updated Figure 4 to make it easier to read.
> Yes, single task means average performance over the tasks when trained individually.
>
> Reviewer Comment: "In eq.(3) Choosing \sigma_i for a task-specific permutation of the network is a bit confusing, since it could be thought of as a sigmoid function, I suggest using a different symbol."
>
> Response: Although \sigma is the most standard notation for a permutation, we understand the potential confusion in this context. We have replaced \sigma with \rho to address this issue.

---

### Official Review · AnonReviewer2 · 2017-11-28
**The paper proposes a novel multitask learning method which looks at a soft ordering of a set of layers, in a DNN framework, which is learnt along with the parameters for all tasks jointly. The paper is well written and details have been given. The experiments are slightly limited but fairly convincing.**

**Rating:** 7
**Confidence:** 4

**Review:**

- The paper proposes to learn a soft ordering over a set of layers for multitask learning (MTL) i.e.
  at every step of the forward propagation, each task is free to choose its unique soft (`convex')
  combination of the outputs from all available layers. This idea is novel and interesting.
- The learning of such soft combination is done jointly while learning the tasks and is not set
  manually cf. setting permutations of a fixed number of layer per task
- The empirical evaluation is done on intuitively related, superficially unrelated, and a real world
  task. The first three results are on small datasets/tasks, O(10) feature dimensions, and number of
  tasks and O(1000) images; (i) distinguish two MNIST digits, (ii) 10 UCI tasks with feature sizes
  4--30 and number of classes 2--10, (iii) 50 different character recognition on Omniglot dataset.
  The last task is real world -- 40 attribute classification on the CelebA face dataset of 200K
  images. While the first three tasks are smaller proof of concept, the last task could have been
  more convincing if near state-of-the-art methods were used. The authors use a Resnet-50 which is a
  smaller and lesser performing model, they do mention that benefits are expected to be
  complimentary to say larger model, but in general it becomes harder to improve strong models.
  While this does not significantly dilute the message, it would have made it much more convincing
  if results were given with stronger networks.
- The results are otherwise convincing and clear improvements are shown with the proposed method.
- The number of layers over which soft ordering was tested was fixed however. It would be
  interesting to see what would the method learn if the number of layers was explicitly set to be
  large and an identity layer was put as one of the option. In that case the soft ordering could
  actually learn the optimal depth as well, repeating identity layer beyond the option number of
  layers.

Overall, the paper presents a novel idea, which is well motivated and clearly presented. The
empirical validation, while being limited in some aspects, is largely convincing.

---

> ### Author Response · Authors · 2018-01-03
> **Response to AnonReviewer2**
>
> Reviewer Comment: "The authors use a Resnet-50 which is a smaller and lesser performing model, they do mention that benefits are expected to be complimentary to say larger model, but in general it becomes harder to improve strong models. While this does not significantly dilute the message, it would have made it much more convincing if results were given with stronger networks."
>
> Response: The computational requirements of a larger model were prohibitive for the experiments in this paper, but we plan to use a stronger model for more complex applications in the future.
>
> Reviewer Comment: "It would be interesting to see what would the method learn if the number of layers was explicitly set to be large and an identity layer was put as one of the option. In that case the soft ordering could actually learn the optimal depth as well, repeating identity layer beyond the option number of layers."
>
> Response: This is an interesting area of future work. So far, based on this suggestion, we have tested the idea of adding an identity layer in the CelebA domain. Including the identity layer does improve performance somewhat, and the identity layer sees increased usage in most contexts. The identity layer creates more consistency across contexts, which can make it easier for soft ordering layers to handle each context effectively. For more details, the results have been added to Section 4.4 in the newly uploaded version of the paper (revised according to these reviews).
>
> We are currently working on the larger topic of developing methods that optimize the size and design of soft ordering models. In one such experiment, the set of modules to be used in the soft ordering framework is designed automatically. The modules can be heterogeneous and consist of multiple layers. Interestingly, in the final set of optimized modules, one module is always a single convolutional layer with no nonlinearity, suggesting that including such pass-through structure, similar to including the identity layer, is important to scaling performance. The initial results are indeed promising, but it is a big topic and will take more time to study.

---

### Author Response · Authors · 2018-01-03
**New Revision Uploaded**

A new revision has been uploaded with changes based on reviewer feedback. For more information about the changes, please see the responses to each reviewer's comments.

---

### Decision · Program_Chairs · 2018-01-29
**ICLR 2018 Conference Acceptance Decision**

**Decision:**

Accept (Poster)

**Comment:**

PROS:
1. Clear, interesting idea.
2. Largely convincing evaluation
3. Good writing

CONS:
1. The model used in the evaluation is a Resnet-50 and could have been more convincing with a more SOTA model.
2. There is some concern about the whether the comparison of results (fig 6c) is really apples to apples.